# Monitoring Apricot (*Prunus armeniaca* L.) Ripening Progression through Candidate Gene Expression Analysis

**DOI:** 10.3390/ijms23094575

**Published:** 2022-04-20

**Authors:** Beatriz E. García-Gómez, Juan A. Salazar, Jose A. Egea, Manuel Rubio, Pedro Martínez-Gómez, David Ruiz

**Affiliations:** Department of Plant Breeding, Centro de Edafología y Biología Aplicada del Segura-Consejo Superior de Investigaciones Científicas (CEBAS-CSIC), 30100 Espinardo, Spain; beatriz.garcia@cragenomica.es (B.E.G.-G.); jasalazar@cebas.csic.es (J.A.S.); jaegea@cebas.csic.es (J.A.E.); mrubio@cebas.csic.es (M.R.); druiz@cebas.csic.es (D.R.)

**Keywords:** *Prunus armeniaca* L., fruit quality, ripening, transcriptomics, metabolomics, breeding, predictive models, multiple linear regression

## Abstract

This study aimed at the monitoring of the apricot (*Prunus armeniaca* L.) ripening progression through the expression analysis of 25 genes related to fruit quality traits in nine cultivars with great differences in fruit color and ripening date. The level of pigment compounds, such as anthocyanins and carotenoids, is a key factor in food taste, and is responsible for the reddish blush color or orange skin and flesh color in apricot fruit, which are desirable quality traits in apricot breeding programs. The construction of multiple linear regression models to predict anthocyanins and carotenoids content from gene expression allows us to evaluate which genes have the strongest influence over fruit color, as these candidate genes are key during biosynthetic pathways or gene expression regulation, and are responsible for the final fruit phenotype. We propose the gene *CHS* as the main predictor for anthocyanins content, *CCD4* and *ZDS* for carotenoids content, and *LOX2* and *MADS-box* for the beginning and end of the ripening process in apricot fruit. All these genes could be applied as RNA markers to monitoring the ripening stage and estimate the anthocyanins and carotenoids content in apricot fruit during the ripening process.

## 1. Introduction

Apricot fruit displays great diversity concerning quality traits as a result of its wide genetic background. This variability has allowed the selection of apricot varieties with interesting phenotypes and to develop new apricot cultivars with improved qualities such as enriched with nutrients, flavor compounds, fiber, vitamins and antioxidants [1]. This has numerous benefits from the point of view of the consumer, it has a visually attractive color, appealing apricot aroma, delightful taste and has health-promoting and protective properties due to the presence of some secondary metabolites like carotenoids or anthocyanins that are supplied in the human diet through the consumption of the edible portion of the apricot fruit [2,3,4]. To reach this high level of diversity, hybridization has been performed between contrasting cultivars, and recombinant genotypes have been obtained to examine their phenotype–genotype correlation for the study of specific ripening-related genes linked to its characteristic phenotypes [5,6,7,8,9].

The achievement of high fruit quality is one of the main objectives proposed in breeding programs. The ripening process is an essential step and the final stage during fruit development, which gives rise to ripe fruit with the acquisition of fruit quality traits due to physiological differentiation and biochemical compound accumulation. All these changes are the result of a coordinated modulation of the gene expression network regulated by complex and interrelated mechanisms affected by internal and external factors [10,11].

A better understanding of the molecular biology and the application of new genetic engineering technologies are proposed as the best strategies for improving fruit quality [12]. Currently, the mechanisms applicable to the regulation of the ripening process are early harvesting, control of the storage atmosphere, chemical and hormonal treatments, and genetic selection of slow or late-ripening varieties. Although these approaches are helpful, they are not universally applicable and often result in acceptable products but of low or poor quality [12].

The identification of genes related to the fruit ripening process has resulted not only in a valuable tool to elucidate the direct effects of specific gene products on the ripening process and the acquisition of final phenotype, but also is an opportunity to resolve the regulatory mechanism and gene expression dynamics along the ripening process [13,14]. In apricot, the genes expressed during the ripening process related to aroma [5,15], taste [15,16], ethylene [9,17], carotenoids [9], phenylpropanoids [15], flavonoids [18] and anthocyanins [19] were monitored. The monitoring analysis of apricot firmness loss was mainly performed post-harvest [6,20,21].

On the other hand, the level of pigments such as anthocyanins and carotenoids are responsible for reddish blush color in the skin or orange skin and flesh color in apricot fruit, which are desirable quality traits in apricot breeding programs. Additionally, these compounds supplied to the diet have several beneficial health effects [1,22]. Considering the genetic resources available, apricot is a very attractive species to identify key genes involved in fruit development and the ripening process [14]. Therefore, monitoring the expression of major genes involved in the main active pathways of the ripening process and its phenotype correlation may elucidate the key genes responsible for fruit quality traits in apricot fruit.

The objective of this work is the analysis of the expression dynamics of candidate genes related to fruit color and the ripening progression obtained from QTLs and RNA-Seq results by using qPCR during the fruit ripening process in nine reference apricot cultivars which gathered a large variability in skin color, flesh color and ripening date. This expression analysis will establish a correlation between expression patterns and their specific phenotypes during the ripening process. This valuable information together with the formulation of multiple linear regression models helps to identify the genes responsible for anthocyanin and carotenoid content in apricot fruit during the ripening process, and quantify the effect of their expression over the mentioned contents.

## 2. Results

### 2.1. Phenotypic Evaluation

Statistical descriptive analyses were performed over the phenological (ripening date) and fruit quality traits (fruit weight, skin color, flesh color, soluble solids content (SSC), acidity (pH), anthocyanins and carotenoids content) of nine cultivars ‘Bergeron’, ‘Cebasred’, ‘Currot’, ‘Dorada’, ‘Estrella’, ‘Goldrich’, ‘Moniquí’, ‘Valorange’ and ‘Z 108-38′ at physiological ripening (Table 1). The normality and homoscedasticity hypotheses, as well as the existence of significant differences between the median values, were tested for each quality trait assayed with the Shapiro–Wilk Rank Sum test, Levene test and Kruska–Wallis test, respectively. While homoscedasticity applies for all the samples, the normality assumption is not fulfilled for firmness, chlorophyll *b* content, anthocyanins and carotenoids content, hence, a non-parametric Kruskal–Wallis test was applied to test the existence of differences between the median values for each quality trait assayed in the nine apricot cultivars analyzed. In all cases, except for SSC, chlorophylls *a* and *b*, *p*-values < 0.05 were found.

These phenological and quality traits displayed wide ranges of values, representing their diversity. For carotenoid content, the white cultivar ‘Moniquí’, showed the lowest value of 3.53 ± 1.62 mg/100 g FW, whereas the orange–red cultivar ‘Valorange’, enriched in carotenoids presented a value of 100 ± 4.80 mg/100 g FW, thirty times higher than ‘Moniquí’. For ripening date, cultivars range from the very early cultivar ‘Cebasred’, which ripens on 14th May (134 ± 3.21 Julian days); in contrast to the late-ripening date cultivar ‘Dorada’, which ripens on 27th June (178 ± 2.52 Julian days), more than a month later. Fruit weight also ranges from the small cultivar ‘Currot’, with an average weight of 49.2 ± 6.62 g; while the cultivar ‘Goldrich’ reaches 91.9 ± 1.94 g, almost twice that of ‘Currot’. Skin color ranges from the reddish ‘Cebasred’, with 64 ± 0.62 ° hue value (the lower, the reddish color expressed in ° hue values); while white-light yellow cultivar ’Moniquí’, shows a value of 95.2 ± 3.38 ° hue, without red blush. Anthocyanins content showed a lower concentration in the light-yellow cultivar ‘Currot’, with 0.51 ± 0.64 mg/100 g FW; while the cultivar ‘Valorange’, mainly covered with an intense red blush, had 18.7 ± 1.21 mg/100 g FW, more than thirty times higher than ‘Currot’. From these results, the transgressive cultivars were ‘Moniquí’, with white fruit color and lower carotenoids and anthocyanins content; and ‘Cebasred’, with reddish skin color, higher anthocyanins and carotenoids content, and a very early ripening date (Table 1).

Boxplots for each quality trait assayed in the nine cultivars ‘Bergeron’, ‘Cebasred’, ‘Currot’, ‘Dorada’, ‘Estrella’, ‘Goldrich’, ‘Moniquí’, ‘Valorange’ and ‘Z 108-38’ at physiological ripening show the diverse range of values of quality traits gathered in the selection of this group of apricot cultivars (Figure 1). Different sizes of the boxes denote a lack of homocedasticity on the samples, while non-symmetrical boxes indicate a lack of normality. The main statistical differences between cultivars were found in carotenoids content (*p*-value = 0.001355), followed by ripening date (*p*-value = 0.001417), fruit weight (*p*-value = 0.003072), acidity (*p*-value = 0.003091), skin color (*p*-value = 0.004857), firmness (*p*-value = 0.005879) and anthocyanins content (*p*-value = 0.007814). These differences confirm the diversity of the considered varieties both in phenology and fruit quality traits.

Variables-PCA plot (Figure 2) represents the contribution of each variable to PCA dimensions 1 and 2, with a 34.6% and 20.5% of variance explanation, respectively. The higher contribution of the variables skin color, flesh color and carotenoids content was found with dimension 1, which explains 34.6% of the variance. It must be taken into account that hue values above 90 are closer to white, between 80 and 90 to yellow, 75 and 80 to light orange, 70 and 75 to orange, and below 70 the color tends to be more reddish. The variables fruit weight and ripening date display a higher correlation with dimension 2. Firmness and soluble solids content had a very low correlation with dimension 1 or 2, but a higher correlation with dimension 5.

Individuals-PCA plot (Figure 3) showed cultivar distribution samples concerning the main PCA dimensions 1 and 2. The cultivars analyzed displayed a wide range of skin color, flesh color and carotenoids content from the white fruit and lower carotenoids content ‘Z 108-38′ to the orange–red and enriched in carotenoids content ‘Valorange’, distributed in a gradient according to its skin color, flesh color and carotenoids content along dimension 1, from orange–red and orange enriched in carotenoids cultivars in the left to white/light yellow and lower levels of carotenoids cultivars in the right. Besides, ripening date trait showed a high contribution to PCA dimension 2, displaying a gradient from the early ripening date ‘Cebasred’, at the bottom left side; to the late-ripening date ‘Dorada’, located at the top right.

Multivariate analysis and PCA are useful tools for cultivar description and characterization of the main differences found between the phenological and fruit quality traits analyzed. The selection of assayed cultivars was based on analyzing the apricot cultivars with major differences in fruit color. The statistical results obtained support the hypothesis that these cultivars display a wide range fruit color and ripening date, an ideal scenario to find differences in the relative gene expression of candidate genes involved in the biosynthetic pathways responsible for fruit color.

Focusing on anthocyanins and carotenoids content during all the ripening stages analyzed (Figure 4), as expected, the highest anthocyanins and carotenoids content during ripening process were found in the cultivars with orange–red and orange fruit color and intense red blush. Anthocyanins content were mainly detected at the end of the ripening process, while carotenoids content gradually increased throughout the entire maturation process in all the cultivars assayed, except for light yellow cultivars which showed their maximum carotenoids content during color change and decreased at physiological ripening.

### 2.2. Monitoring Gene Expression of Candidate Genes

For monitoring the gene expression of candidate genes, the analyses were focused on previous results obtained in RNA-Seq analysis, where candidate genes related to pigment content (anthocyanins and carotenoids) were proposed. Besides, the quality traits which showed the highest differences were also considered. In both cases, the most significant differences were related to fruit color, a descriptor correlated with anthocyanins and carotenoids content, responsible for skin and flesh color in apricot fruit.

The candidate fruit quality-related genes were selected from QTLs linked to quality traits (*NAC* and *MYB10*) and differentially expressed in RNA-Seq (*LOX2*, *GST*, *DXS*, *bHLH*, *bZIP*, *MADS-box*, *CAD1*, *CtrL*, *WD40*, *C4H*, *CL4*, *CHS*, *MATE*, *UFGT*, *WRKY*, *F3′5′H*, *EDR1* and *psbP1*).

The selected genes were mainly involved in anthocyanins biosynthesis as *4CL*, *ANS*, *bHLH*, *C4H*, *CAD1, CHS, DFR, F3′5′H, MYB10, UFGT* and *WD40* (Figure 5); carotenoids biosynthesis as *CCD1*, *CCD4, CrtL, DXS* and *ZDS* (Figure 6); metabolites transporters as *GST* and *MATE* (Figure 7); and others genes involved in the regulation of the ripening progress as *bZIP, EDR1, LOX2, MADS-box, NAC, psbP1* and *WRKY* (Figure 8). These genes were analyzed by qRT-PCR in nine apricot cultivars ‘Bergeron’, ‘Cebasred’, ‘Currot’, ‘Dorada’, ‘Estrella’, ‘Goldrich’, ‘Moniquí’, ‘Valorange’ and ‘Z108-38′ at three ripening stages: green fruit (Sample A), during color change (Sample B) and at physiological ripening (Sample C).

Most of the genes involved in anthocyanins biosynthesis (Figure 5) were found up-regulated in orange–red, orange and yellow fruit color cultivars, whose expression increases during the ripening process, reaching the maximum expression at physiological ripening in most cases. ‘Cebasred’, an ‘Orange Red’ cultivars mainly covered by an intense red blush was the cultivar with the highest expression of genes *ANS, C4H, UFGT* and *WD40,* increasing the expression the ripening process. The gene *CHS* showed its maximum expression in the orange–red and orange cultivars ‘Cebasred’ and ‘Goldrich’. The orange cultivar ‘Bergeron’ showed the highest expression of the transcription factor *MYB10* and the gene *4CL*. The gene *F3′5′H*, responsible for anthocyanin hydroxylation, was highly up-regulated in the yellow fruit cultivar ‘Dorada’, less expressed in the other orange–red or orange cultivars, and completely silenced in the light yellow and white ones. On the other hand, the transcription factor *bHLH* appeared up-regulated in light yellow cultivars such as ‘Currot’ and ‘Moniquí’. We found the maximum gene expression of *UFGT* in the contrasted fruit color cultivars ‘Currot’ and ‘Cebasred’.

Furthermore, candidate genes involved in carotenoids biosynthesis and degradation (Figure 6) were monitored. The gene *DXS*, a precursor of carotenoids compounds, showed increased expression during the ripening process, reaching its maximum at physiological ripening. The highest expression of *DXS* at physiological ripening was detected in the cultivar ‘Currot’, followed by ‘Cebasred’ and ‘Valorange’. As expected, the genes *ZDS* and *CrtL*, involved in carotenoids biosynthesis were found up-regulated in orange–red cultivars, increasing their expression during the ripening process. Besides, the light yellow cultivar ‘Currot’ showed an overexpression in the gene *ZDS*. The genes responsible for carotenoids degradation were included in the analysis due to their importance for carotenoids metabolism. In this case, *CCD4* was highly up-regulated in the light yellow cultivars ‘Currot’ and ‘Moniquí’. No clear pattern was identified in the gene expression of *CCD1.*

Most of the secondary metabolites (e.g., anthocyanins) are accumulated in the vacuole where the acid environment conditions the color displayed by molecules depending on their structure and residues. Two metabolite transporters into the vacuole were analyzed (Figure 7). The genes *GST* and *MATE* were proposed as candidate metabolite transporters from the cytoplasm to the apoplast. Both genes were found up-regulated in light yellow and white cultivars ‘Currot’, ‘Moniquí’ and ‘Z 108-38′.

Finally, the analysis included some transcription factors and possible regulators of the ripening process (Figure 8) found differentially expressed in the RNA-Seq analysis. Two genes with contrasted expression patterns during fruit ripening were identified. The gene *LOX2* seems to be up-regulated at the beginning and its expression decreases during the ripening process. On the other hand, the gene *MADS-box* seems to have the opposite expression pattern, i.e., its expression increased during the ripening process, reaching its maximum at physiological ripening. No clear pattern related to fruit color and ripening stage was identified in the gene expression of *bZIP, EDR1, NAC* and *WRKY*.

### 2.3. Correlation between Gene Expression of Assayed Genes

Pearson correlation coefficients among relative gene expression in different ripening stages are shown as a correlation plot in Figure 9. Most of the correlations found had positive values due the increased expression of most of the genes analyzed as the result of the activation of the biosynthetic pathway in which they are involved.

Pearson correlations over 0.7 were plotted in a correlation network of relative gene expression (Figure 10). The Pearson correlation between gene expression shows higher positive correlations between the genes mainly involved in anthocyanins (*C4H-CHS*) and carotenoids (*CrtL-ZDS*) biosynthesis, and between some transcription factors and genes (*WRKY-CCD1* and *MADS-box-bZIP*). Only significant positive correlations were found.

### 2.4. Multiple Linear Regression Model

Multiple linear regression (MLR), as an extension of the simple linear regression model, is a powerful technique used to predict the value of a response variable (*y*) from the values of two or more variables (**x** = {*x*_1_, *x*_2_, …, *x*_n_}), also called predictors. Due to the polygenic nature of these fruit quality traits, this approach is more appropriate than a simple linear regression model, where each gene is analyzed individually. Two MLR models were built to estimate the content of anthocyanins and carotenoids in every ripening stage assayed, where the dependent variable was the cultivars phenotype (anthocyanins and carotenoids content), and the independent variables were the relative gene expression of genes involved in anthocyanins and carotenoids biosynthesis, metabolites transporters and ripening process regulators.

#### 2.4.1. Anthocyanins

For the MLR model to predict anthocyanins content, included first in the model were the genes involved in ester biosynthesis (*LOX2*), flavonoid biosynthesis (*CAD1*), anthocyanin biosynthesis (*C4H, 4CL, CHS, F3′5′H, DFR, ANS* and *UFGT*), metabolites transporters (*GST* and *MATE*), ethylene perception (*EDR1*) and transcription factors (*MYB10, bHLH, WD40, bZIP, MADS-box, WRKY* and *NAC*). After a stepwise variable selection method based on the Akaike Information Criterion (AIC) the genes *ANS, WD40, C4H, MYB10, CHS, F3′5′H, WRKY, bHLH, 4CL, LOX2* and *MADS-box* were selected. Finally, the following model was obtained to estimate the anthocyanins content (mg/100 g FW):

Anthocyanins = 7.24 + 2.60 ANS − 3.28 WD40 − 3.82 C4H − 0.55 MYB10 + 5.64 CHS − 2.24 F3′ 5′H − 5.11 WRKY + 3.57 NAC − 9.27 bHLH − 4.13 4CL − 4.88 LOX2 + 12.22 MADSbox.

Our model, where each abbreviation represents the relative expression of its corresponding gene, had a residual standard error of 2.909 on 14 degrees of freedom, adjusted R^2^ = 0.6745 and *p*-value = 0.00176, showing statistical significance.

Even after the stepwise model selection, most of the predictors had low statistical significance, which is an indication that they might not contribute to the model or their contribution is very low, such as *ANS, WD40, C4H* and *MYB10*. The highest statistical significance in the anthocyanins model was observed for the genes *MADS-box, CHS, NAC, bHLH* and *4CL*.

In order to validate the statistical hypotheses of the MLR model, the normality and homoscedasticity of the residuals were evaluated by a normal Q–Q plot and residuals vs. fitted values (Appendix A). The Q–Q plot shows that the residuals were normally distributed as the data showed a linear pattern. The residuals vs. fitted values plot showed a pattern, with positive values in all the residuals for low fitted values and negative ones for high fitted values, following a straight line. Therefore, the model should be managed with caution if used for inference purposes.

#### 2.4.2. Carotenoids

For the MLR model to predict carotenoids content we first included in the model the genes involved in ester biosynthesis (*LOX2*), carotenoids metabolism (*CrtL, CCD1, CCD4, DXS* and *ZDS*), metabolites transporters (*GST* and *MATE*), ethylene perception (*EDR1*) and transcription factors (*bZIP, MADS*-box and *NAC*). After the same AIC-based stepwise variable selection method used for the anthocyanin model, the *genes CCD4, CCD1, ZDS, bZIP, EDR1, MADS-box, NAC, LOX2* and *CrtL* were selected. Finally, the following model was obtained to estimate the carotenoids content (mg/100 g FW):

Carotenoids = 67.75 − 72.94 CCD4 − 27.30 CCD1 + 47.00 ZDS + 74.20 bZIP − 33.07 EDR1 − 34.31 MADSbox − 45.96 LOX2 + 9.09 NAC + 1.23 CrtL.

Our model, where each abbreviation represents the relative expression of its corresponding gene, had a residual standard error of 17.8 on 17 degrees of freedom, adjusted R^2^ = 0.6794 and *p*-value of 0.0002856. The lower statistical significance was obtained for the genes *CCD1, CrtL, NAC* and *MADS-box*, with a *p*-value > 0.05. The predictors with the highest statistical significance were *ZDS*, with a *p*-value < 0.05; *bZIP, EDR2* and *LOX2*, with a *p*-value < 0.01; and the most significant *CCD4*, with a *p*-value < 0.001.

Like in the case of anthocyanins content, normality and homoscedasticity hypotheses of the MLR model were evaluated by a normal Q–Q plot and residuals vs. fitted values plot (Appendix A). The Q–Q plot revealed a normal distribution of the residuals. The residuals vs. fitted values plot did not present any pattern, which means that the homoscedasticity hypothesis is also fulfilled. Therefore, this model could be used not only for prediction but also for inference purposes.

#### 2.4.3. Anthocyanins and Carotenoids MLR Models Gene Correlation Prospection

When performing an MLR model, the model selection removes those predictors that either do not influence the response or are correlated with other predictors. If the latter occurs, the model is still valid for predicting, but it would not be capturing all the expressions that influence the response, keeping only the most significant ones and discarding others that are present but are correlated with the first ones and do not enter the model. In this sense, the model does not give an idea at first glance of all the genes that influence the metabolite content. Hence, to analyze the correlation between the genes included in the MLR models for the prediction of anthocyanins and carotenoids content, a PCA of the relative gene expression was performed, including all the genes initially considered in both MLR models to identify those predictors correlated with other predictors or with no influence on anthocyanins or carotenoids content.

In the PCA for anthocyanins content MLR model, dimensions 1 and 2 had 31.5% and 14.6% of the variance explanation, respectively. From the variables-PCA plot for the relative expression of the genes involved in anthocyanins content (Figure 11), the higher variable contribution to dimensions 1 and 2 was due to the genes *MADs-box, C4H* and *CHS*. The relative expression of the genes *MADS-box-bZIP* and *CL4-NAC* perfectly overlapped when represented dimensions 1 and 2. The variables *LOX2, F3′5′H* and *GST* had very low or no contribution to dimension 1, while *UFGT* and *WD40* had a meager contribution to dimension 2. Other high contributions were found for the relative expression of gene *LOX2* to dimension 3, *F3′5′H* to dimension 5 and *GST* to dimension 4.

When analyzing the PCA for carotenoids content MLR model, the dimensions 1 and 2 capture 37.1% and 17.5% of the variance explanation, respectively. In the variables-PCA plot (Figure 12), the higher variable contribution to dimensions 1 and 2 were due to the genes *MADS-box, ZDS* and *CCD4*. The relative expression of the genes *CCD4-GST* and *CCD1-CrtL* overlapped when represented dimensions 1 and 2. The genes *LOX2, GST* and *NAC* had no contribution to dimension 1; while the genes *CrtL, CCD1* and *DXS* had no contribution to dimension 2. Other high contributions were found for the relative expression of gene *LOX2* to dimension 4, *GST* to dimension 3 and *NAC* to dimension 5.

These results corroborated the significant contribution of the *CHS* gene in the MLR model for anthocyanins content, and the *CCD4* gene in the MLR model for carotenoids content. In both PCA analyses, *MADS-box* showed a higher contribution to the main dimension while *LOX2* and *GST* had the lower contributions, which could be due to their important co-expression but not direct effect into anthocyanins or carotenoids content.

## 3. Discussion

The attempt to generate a model based on the expression of genes involved in a metabolic process responsible for a specific phenotype could be helpful to validate these candidate genes and predict interesting phenotypes. For this purpose, two MLR models were built to identify the key genes involved in the anthocyanin and carotenoid biosynthesis, followed by a relative gene expression PCA to corroborate the weight of candidate genes into both MLR models.

### 3.1. Multiple Linear Regression Model for Anthocyanins Content

From the point of view of molecular breeding, well-known features of the anthocyanin pigments and other phenylpropanoids and flavonoids compounds are of great interest for increasing the attractiveness and health benefits of fruits [23,24,25,26,27]. Changes in anthocyanins accumulation affect skin color during the ripening process, being responsible for the blush red coloration in apricot skin [4,28]. The expression of anthocyanins biosynthesis genes in apricot species correlates with anthocyanin accumulation and red coloration in the skin fruit [28]. Besides apricot [19], anthocyanin biosynthetic pathway was described in related species such as peach [29,30,31], Japanese plum [32,33,34], Japanese apricot [35], apple [36,37,38] and pear [39].

Several genes related to phenylpropanoids, flavonoids and anthocyanins biosynthesis were analyzed to build a MLR model for anthocyanins content in apricot fruit during the ripening process. From the phenylpropanoids biosynthesis pathway, are considered the gene expression of cinnamyl-alcohol dehydrogenase (CAD1), that transforms cinnamyl alcohol into cinnamaldehyde as the precursor of phenylpropanoids biosynthesis pathway; trans-cinnamate 4-monooxygenase (C4H), responsible for hydroxylation of cinnamic acid to p-coumaric acid; and a 4-coumarate-CoA ligase 1-like (4CL) activated p-coumaric acid into its thioester or p-coumaroyl-CoA. Following with the flavonoid biosynthesis pathway, is included the gene for chalcone/stilbene synthase (CHS), catalyzing the reaction from p-coumaroyl-CoA to naringenin chalcone. Other genes of interest are flavonoid 3′5′-hydroxylase 1-like (F3′5′H) as the enzyme that hydroxylates anthocyanins yielding dihydroflavonols. Dihydroflavonol reductase (DFR) catalyzes the reduction from dihydroflavonols to leucoanthocyanidins; anthocyanidin synthase (ANS) leads anthocyanidin followed by the action of UDP flavonoid 3-O-glucosyltransferase (UFGT), whose activity is required to catalyze the formation of stable glycosylated anthocyanin, thus, acquiring their final structure and color. *C4H, 4CL, CHS*, *F3′5′H, DFR* and *UFGT* were found differentially expressed in RNA-Seq experiments contrasting two apricots genotypes which differ in skin and flesh color. It has been shown that in the Rosaceae family, *CHS* and *F3′5′H* are early genes and have a coordinated expression pattern, reaching their maximum expression in the first stage of fruit development [40]. The late gene *ANS* has its peak of maximum expression at the physiological ripening stage, especially in tissues that accumulate anthocyanin in the skin, as is the case of apricot fruit [4].

In peach, in agreement with the pattern of anthocyanin accumulation during fruit ripening, *CHS, F3H* and *DFR*, were expressed at higher levels in skin and mesocarp around the stone and at lower, sometimes almost at background levels, in the mesocarp. A positive correlation between gene expression and anthocyanins content was found for *CHS, F3H, DFR* and *ANS* [31]. ANS protein was also identified, increasing in abundance during the transition from unripe to the ripe stage [41]. In Japanese plum, the EBGs genes *CHS, F3H* and *DFR* showed a peak of expression in S1 both in skin and flesh, followed by a second peak expression in S3 of *CHS* in red cultivars, while they showed a basal expression pattern in the yellow cultivars. On the other hand, the expression of *DFR* was uniform in the flesh of all cultivars, high in S1, and reduced to the basal levels in the succeeding stages. Late gene *ANS* showed expression peaks in the S3 and S4 stage in the skin of red cultivars. All tissues without anthocyanin accumulation at all development stages show a minimal expression of *ANS*, while pigmented tissues increased expression, suggesting this gene is expressed in a coordinated manner during the changes of fruit color. The major correlation between anthocyanins content and gene expression in skin and flesh was found with *ANS* expression [33]. Studies in green apples and yellow cherries have found that early genes *4CL, CHS, F3H* and *DFR* were not expressed, or were in meager amounts, suggesting their importance during the synthesis of anthocyanins [37,42]. In addition, in Japanese plum, in cultivars with red skin, accumulation of anthocyanins began in S2 and continued in S3 and S4, contrary to what was observed for proanthocyanidins, which accumulated mainly in S1 in cultivars with red skins [33].

As a result of monitoring the relative expression of *ANS* and *CHS*, higher relative expressions in the cultivars with orange–red and orange fruit color ‘Cebasred’, ‘Estrella’ and ‘Valorange’ were found, which may explain the accumulation of anthocyanins content at physiological ripening in these cultivars. *F3′5′H* shows higher relative expression in orange and yellow cultivars ‘Dorada’ and ‘Bergeron’, while no expression was detected in any light yellow or white cultivars ‘Currot’, ‘Moniquí’ and ‘Z 108-38′. *4CL* shows its higher relative expression in orange cultivar ‘Bergeron’ at physiological ripening. *C4H* shows its higher relative expression in the reddish cultivar ‘Cebasred’ at physiological ripening. This suggests that *C4H*, *4CL, F3′5′H, ANS* and *CHS* are the key regulation points in the flavonoid pathway, which participated in the incorporation of phenolic compounds to the anthocyanins biosynthesis pathway; and its regulation could be crucial for anthocyanins accumulation during the fruit ripening process in apricot [43].

The spatial and temporal distribution of anthocyanin biosynthesis in fruits is due to the regulatory mechanism behind the transcription factor complexes composed of MYB-bHLH-WD40, a MBW transcriptional complex. The biological significance of this sophisticated regulatory mechanism is the plant requirements to produce anthocyanins in the correct spatial location, time and at the appropriate levels to fulfil the differing demands of plant development and environmental response. R2R3-MYB genes, such as *MYB10*, have been demonstrated to be responsible for the accumulation of anthocyanins in apricot [19,40], and several fruit crops, including grape [44], apple [36,45], peach [46,47,48], Japanese apricot [35] and Japanese plum [49,50].

Anthocyanin biosynthesis in apricot is seen to be regulated by the complex transcription factor MBW, where the increasing expression of *PaMYB10* during the fruit ripening process was described [19]. The inclusion of *MYB10*, *WD40* and *bHLH* genes in the MLR model and the existence of a correlation of gene co-expression between C4H and WD40 genes may result in a regulatory mechanism which controls anthocyanin biosynthesis in apricot fruit. From these genes, *bHLH* has the major estimate coefficient and lowest *p*-value. In the transcription regulation of anthocyanins, MYB10 elicits a critical effect on combining and activating transcription. WD40 protein promotes the interaction between proteins. While bHLH stabilizes the MYB protein structure, promotes its transcript activation and determines its tissue specificity [51]. On the other hand, some MYB transcription factors do not need the assistance of bHLH to regulate the transcription of anthocyanin genes. In the correlation plot between gene expression, *bHLH* shows a higher correlation with *CHS*, possibly related to the induction on its expression by joining it promoter [52].

In peach, the expression of *MYB10.1* and *MYB10.3* was detectable in mesocarp and skin, being highest in mesocarp around the stone (Cs), at slightly lower levels in the skin and at minimum in mesocarp, where bHLH3 was found to have an expression profile correlating with anthocyanin accumulation and higher expression of *DFR*, *CHS* and *UFGT*. Functional analysis of peach *MYB10* and *bHLH* genes by agrobacterium-mediated transient transactivation assays in tobacco showed that the best MYB10.1 partner was bHLH3, mediating the expression of *CHS*, *DFR* and *UFGT* in agro-infiltrated tobacco leaves and peach fruits [46]. In Japanese plum, a sustained increase in expression of *PsMYB10* began in S2 in the skin of all red cultivars, and continued until S4; it showed the highest positive correlation with anthocyanin accumulation, *ANS* and *UFGT* gene expression, suggesting a putative function of *PsMYB10* in the regulation of transcription during anthocyanin biosynthesis. On the other hand, there exists a significant negative correlation between anthocyanin accumulation, *ANS* and *UFGT* gene expression and the highest expression of *PsMYB1* in all yellow tissues [33]. Besides, the genomic variability of *PsMYB10.1* were found highly associated with anthocyanin and anthocyanin-less skin [50]. In kiwi fruit, it was described that the MBW complex is responsible for regulating anthocyanin biosynthesis rather than having a direct effect on anthocyanins content [53].

In our MLR model, only the expression of gene *bHLH* has a statistical significance, and was found up-regulated in light yellow and white cultivars. Neither statistical significance nor relative expression patterns were found for *MYB10* and *WD40*. One explanation for that result could be the influence of environmental conditions for the development of red blush color in the skin fruit in apricot. Anthocyanin biosynthesis is enhanced by light and temperature, and MYB TFs appears to be the primary determinant of fruit pigmentation in response to light [38,54,55]. In apricot fruit, only the side exposed to sunlight develops red blush [28]. The regulatory effect of light has also been described in peach [47], pear [56] and apple [57,58,59]. In apple, the exposition to the light of the shaded side of fruit leads to the up-regulation of *CHS, CHI, F3H, DFR, ANS* and *UFGT*, which is followed by an increase in the anthocyanins content and red coloration [58]. Another level of regulation of the ripening process is based on epigenetic mechanisms, such as promoter methylation of R2R3 MYB genes, described as a crucial role in the regulation of anthocyanin accumulation in fruits like apple [60] and pear [39].

Besides the MBW complex described above, other TFs and regulatory effectors such as PHYTOCHROME-INTERACTING FACTOR 3 (PIF3) [61], ELONGATED HYPOCOTYL 5 (HY5) [62], COP1 [63], WRKY TFs [64], TRANSPARENT TESTA 1 (TT1, WIP domain) [65], TRANSPARENT TESTA16 (TT16, MADS-domain) [66], NAC TFs (NAM (for no apical meristem), ATAF1 and −2, and CUC2 (for cup-shaped cotyledon)) [67], JASMONATE ZIM-domain (JAZ) [68] and SQUAMOSA PROMOTER BINDING PROTEIN-LIKE (SP [69] have also been reported in Arabidopsis to affect anthocyanin biosynthesis, most of them involved in light-regulated transcriptional activation. Some of these effectors, such as activator PpNAC1 TF and repressor PpSPL1, show an interaction with the MBW complex, as transcriptional regulators of the genes encoding anthocyanin biosynthetic pathway enzymes in blood-fleshed peach [31]. In Petunia, WRKY TF controls the flavonoid pigment pathway through its relationship with the MBW complex, suggesting a new regulatory model where MBW bind WRKY modifying the transcriptional activity of genes encoding transporters of pigment precursors into the vacuole, as the candidate metabolite transporters MATE or GST [70]. The results obtained from monitored *WRKY*, *NAC* and *MYB10* seem to be up-regulated at physiological ripening in the cultivar ‘Valorange’, with an intense red blush. *MATE* and *GST* were mainly found up-regulated in light yellow cultivars, which do not correlate with their expected function as anthocyanins transporters into the vacuole.

In ripe apricots, anthocyanins concentration increased in the skin, reached a maximum value and decreased toward the end of the maturation phase. There are two possible explanations for the anthocyanins decrease in the skin of ripe apricots. The first involves the degradation of the anthocyanins, molecules known to be unstable in weakly acidic conditions. A second explanation could be associated with a dilution effect induced by fruit growth due to fruit growth being faster than anthocyanin biosynthesis and accumulation. Lastly, the sampling method should be considered, the peeling became difficult in ripe apricots and yielded two consequences: inclusion of flesh particles in the skin preparation, and longer duration of skin preparation allowing increased oxidation of polyphenols [28]. Considering the pericarp, as the edible part of the fruit, was used to assess the anthocyanins content as a beneficial compound for human health, a dilution effect may be occurring, leading to an underestimation of the real anthocyanins content.

In the MLR model, the genes involved in the anthocyanin biosynthesis with the highest estimate coefficient and lowest *p*-value were *CHS* and *4CL*. The expression dynamics of both genes showed the higher expression of these genes in apricot fruits with orange–red and orange fruit color as ‘Cebasred’, ‘Estrella’, ‘Valorange’, ‘Goldrich’ and ‘Bergeron’ at physiological ripening. These results correlated with the expected phenotype with increased anthocyanins content. Moreover, it should be emphasized that in the diagnostic plots, to check that the model works well with the data, the residuals vs. fitted values plot shows how the model captures non-linear relationships. This is a good example to illustrate that the regression coefficients and statistical descriptors, such as *p*-values or R^2^, are not enough to evaluate how appropriate a selected model is for analyzing data and for prediction purposes. This non-linear relationship may be due to the differential expression of anthocyanins in the apricot skin, the low sensibility of the method used for quantifying anthocyanins content, the presence of interactions between the predictors or the lack of inclusion of important predictors during the model selection step.

As mentioned above, this model did not fulfil the homoscedasticity hypothesis. Although it was reformulated including all possible interactions between the genes described above, it was not possible to find any model fulfilling the hypothesis. Therefore, the MLR model for anthocyanins content prediction must be used with caution. To complement the result obtained with MLR model, the relative expression PCA of all the genes initially included in the analysis could help the selection of all the genes that influence the metabolite content, including those predictors that were removed from the MLR model as a result of not influencing the response or because they were correlated with other predictors. In our case, only the relative gene expression of *CHS* was found significant both in the MLR model and PCA.

Taking into consideration the multiples levels of anthocyanins biosynthesis regulation, it made it harder to develop a MLR model only based on the relative gene expression of candidate genes. Hence, environment effects, epigenetic gene expression control and the origin of the tissue sampled should be considered for more precise model construction in the future. Finally, *CHS* is proposed as a candidate gene for predicting the presence of reddish blush and anthocyanins content in apricot fruit during the ripening process.

### 3.2. Multiple Linear Regression Model for Carotenoids Content

In the edible portion of apricot, *β*-carotene is the main pigment followed by *β*-cryptoxanthin and *γ*-carotene [3]. In apricot, the carotenoids content is highly correlated with skin and flesh color [71] and it is possible to determine the content of carotenoids, *β*-carotene and provitamin A based on a colorimeter measure of the flesh and skin color of the edible portion [72]. The carotenoid biosynthesis pathway was described in apricot [14], and related species such as peach [73,74,75], Japanese apricot [76] and Japanese plum [77]. In apricots, the skin is consumed as an edible portion because it is the enriched part of the apricot, with its carotenoids content 2–3 times higher than in flesh [3].

The accumulation of carotenoids at the end of the ripening process is not only due to the induction by ethylene of the genes involved in carotenoids biosynthesis. There are two different regulatory mechanisms postulated. One is focused on carotenoid sink capacity, and the other is focused on the carotenoids degradation [78]. Carotenoids content and carotenogenic gene expression are regulated by ethylene in apricot. Several genes associated with apricot isoprenoids biosynthesis are under the control of ethylene during the transition from stages S3 to S4 [11]. As described by Marty et al. [9], the gene expression of the carotenoid pathway was analyzed as a function of ethylene production in two color-contrasted apricot varieties, ‘Goldrich’ and ‘Moniquí’. Fruits from ‘Goldrich’ were orange, while ‘Moniquí’ fruits were white. Biochemical analysis showed that ‘Goldrich’ accumulated precursors of the uncolored carotenoids, phytoene and phytofluene, and the colored carotenoid, *β*-carotene; while ‘Moniquí’ fruits only accumulated phytoene and phytofluene but no *β*-carotene. The physiological analysis showed that ethylene production was weaker in ‘Goldrich’ than in ‘Moniquí’. Carotenogenic gene expression (*PSY1*, *PDS*, and *ZDS*) and carotenoid accumulation were measured concerning ethylene production, being initiated in mature green fruits at the onset of the climacteric stage. Results showed systematically stronger expression of carotenogenic genes in white than in orange fruits. Even for the *ZDS* gene involved in the *β*-carotene biosynthesis, ethylene induction of *PSY1* and *PDS* gene expression correspond to *β*-carotene production and it is independent of ethylene. In our study, *ZDS* were found up-regulated in ‘Cebasred’ and ‘Currot’, orange–red and light yellow cultivars, respectively, with the highest and lowest carotenoids content.

Supporting the results obtained by Marty et al. [9], ‘Currot’ has high ethylene production while ‘Cebasred’ has low ethylene production. Therefore, the fruit color due to carotenoids content is independent of carotenogenic gene expression or ethylene production. Besides, the unique gene related to ethylene metabolism included in the MLR model for carotenoids content was *EDR1*, which is homologous to *LeCTR2* based on the phylogenetic analysis performed in Arabidopsis. It has been described as being involved in pathogen response and not in ethylene signaling [79], and no pattern related to carotenoids content or the ripening process was found during gene expression dynamics analysis.

Transcriptomic analysis during the ripening process in peach showed an increase in the expression of several genes involved in the carotenoids pathway, such as *ZDS* and *CrtL*, which may result in a stimulation of *β*-carotene and neoxanthin biosynthesis [11]. In tomato, CtrL was described as a paralog of lycopene β-cyclase (LYCB) and neoxanthin synthase (NXS), yielding new forms of oxidized carotenoids cleaved by carotenoid cleavage dioxygenases (CCD) into apocarotenoids, and degraded by 9-cis-epoxycarotenoid dioxygenase (NCED) into ABA, triggering fruit senescence [80].

In the other level of carotenoids content regulation, as cited previously, carotenoids can be cleaved into volatile apocarotenoids in fruit by the enzymes carotenoid cleavage dioxygenases (CCD). In apricot, a rapid significant increase in CCD activity was found during the fruit development process in both skin and flesh [16]. In peach, two genes were described: CCD1 and CCD4. CCD4 is the gene responsible for flesh color in peach and its expression results in the degradation of carotenoids in white-fleshed genotypes [74], while the yellow color arises as a consequence of its inactivation [73,81]. A CCD4 functional allele is consistently associated with the ancestral white flesh color; on the other hand, the yellow phenotype has originated from at least three independent mutations disrupting CCD4 function, thus preventing carotenoid degradation [82,83].

Monitoring the relative expression of the genes included in the model in the nine cultivars analyzed, for the most interesting genes involved in carotenoids content, *ZDS* and *CrtL*, presented the highest expression in the cultivars with orange–red fruit color ‘Cebasred’, ‘Estrella’ and ‘Valorange’, which may explain the accumulation of carotenoids content at physiological ripening in these cultivars. These genes were also found increased in light yellow and white fruit color cultivars such as ‘Currot’ and ‘Moniquí’, with lower content in carotenoids. But in these cultivars, the gene responsible for carotenoids cleavage *CCD4* was highly up-regulated, which finally explains the lower content of carotenoids and light yellow and white fruit color. No identified pattern was found for *CCD1*.

The cultivars with higher levels of ethylene production were ‘Currot’, ‘Moniquí’, ‘Z 108-38′ and ‘Valorange’. All these cultivars, except for ‘Valorange’, have light yellow or white fruit color. Therefore, it could be concluded that carotenoids content at the end of fruit ripening is independent of the ethylene production and white flesh in apricot is the result of carotenoids cleavage by *CCD4*.

The last gene included in the model was the *bZIP* TF. It its implication in the ripening process has been described in peach [84], apple [85] and strawberry [86]. In peach, its role in the ripening process has been suggested, indicating that it acts as a pacemaker some of the ripening metabolic pathways. Transgenic tomato fruits with constitutive *bZIP* expression show decreased ethylene production and dilating ripening time. Furthermore, *bZIP* affects the expression of *LOX* by decreasing its transcription. Hence, the decrease in the expression of *bZIP* during the ripening process acts as a signal of “end of ripening” in fleshy fruits and maybe it is another candidate gene for monitoring the ripening process [83]. In the nine apricot cultivars assayed, *bZIP* seems to increase its expression during the fruit ripening process in ‘Cebasred’ and ‘Currot’, with the earliest ripening dates, but the highest relative expression of this gene is found in other cultivars such as ‘Valorange’ and ‘Moniquí’ at physiological ripening, and ‘Estrella’ at the beginning of the ripening process.

In the MLR model obtained, the predictor contributing the most to the model was *CCD4*, with the highest statistical significance, followed by *bZIP*, *EDR1* and *LOX2*. Therefore, carotenoids content is more highly correlated with the expression of degradative enzymes such as *CCD4* than with the regulation by ethylene or the expression genes involved in carotenoids biosynthesis. The highest correlation with *LOX2* is probably due to the dependence of the ripening stage in the carotenoids biosynthesis. The MLR fulfils both the normality and homoscedasticity assumptions and could be applied in the prediction of carotenoids content during the ripening process in the apricot fruit. Besides, the relative gene expression PCA corroborate the high contribution of the genes *ZDS* and *CCD4* to carotenoids content.

Finally, the genes *CCD4* and *ZDS* were proposed as candidate genes responsible for carotenoids content. The expression dynamics of *CCD4*, higher in light yellow and white cultivars such as ‘Currot’, ‘Moniquí’ and ‘Z 108-38′ perfectly explain the lower carotenoids content and fruit color at physiological ripening. Thus, the final phenotype is conditioned by carotenoids biosynthesis vs. degradation, being the expression dynamics of *CCD4* and *ZDS,* which are the key genes to determine fruit color and carotenoids content.

### 3.3. Candidate Genes Descriptors for Ripening Stage

The characterization of the fruit ripening stages based on their gene expression depends on the fact that each ripening stage is characterized by a specific increase or decrease in the expression of genes related to the regulation of the ripening process. In both models built for anthocyanins and carotenoids content, the two transcriptions factors *NAC* and *MADS-box*, and the gene *LOX2* showed the highest estimate coefficients and statistical significance. The biosynthesis of anthocyanins and carotenoids is highly correlated with fruit development [87], so it makes sense that genes that are regulators of the fruit ripening process have been included as predictors in both models.

NAC belongs to one of the largest plant transcription factor families, whose members are involved in many developmental processes such as senescence, stress, cell wall formation and embryo development. NAC is linked to ripening date (RD), and its genomic location is nearby to where a QTL was detected with a higher variation (52%) in LG4. Fine mapping of the RD locus in peach identified the candidate gene (*Prupe.4G186800*) encoding a transcription factor of NAC family as a possible causal gene. A sequence variant in a *NAC* candidate gene with a 9 bp insertion was shown to co-segregate with the RD trait and can be used to select early maturity genotypes in peach [88]. The development of a DNA marker based on this sequence polymorphism provides a convenient molecular tool to discriminate early- vs. late-ripening individuals in breeding programs. Besides, *NAC* TF has been proposed to be involved in the regulation of anthocyanin accumulation during blood-orange response to cold exposure [89]. Recently, fruitENCODE epigenome data revealed a transcriptional feedback circuit controlling ethylene-dependent ripening based on methylation marks in specific loci where *NAC* transcription factor is involved through regulating the expression of *ACS* and *ACO* genes. In this epigenetic transcriptional feedback circuit in peach, H3K27me3 tissue-specific methylation in *NAC* and *ACS* were responsible for silencing ripening genes and controlling the progression of the ripening process. In fleshy fruit species, such as peach and tomato, it is a significant evolutionary advantage to use a stable epigenetic mark like H3K27me3 to keep the autocatalytic ripening loop under strict developmental control [90]. In our MLR model and expression gene dynamics analysis, it seems to be up-regulated in orange–red and orange cultivars, but no strong correlations with the ripening date or fruit color were found.

The *MADS-box* gene was included in both models. *MADS-box* genes encode transcription factors that play crucial roles in plant development, especially in flower and fruit development [91,92]. SEPALLATA MADS-box TFs have also been reported to be associated with fruit development in Japanese apricot [91] and anthocyanin accumulation in bilberry [93] and pear [94], respectively. The expression dynamics of this gene during the ripening process may be an indicator of the fruit development stage, increasing at the end of the ripening process in most of the apricot cultivars assayed in this study, except for ‘Estrella’ and ‘Z 108-38′. Thus, *MADS-box* is proposed as candidate gene for monitoring the end of the ripening process.

Another gene included in both MLR models was *LOX2*. In plants, this gene is closely related to fruit ripening and senescence. Additionally, fruit quality traits including fruit firmness, ethylene production and soluble solids content under different storage conditions were affected by the differential expression of *LOX* [95]. In the fatty acid pathway, unsaturated fatty acids linoleic acid (18:2) and linolenic acid (18:3) can be transformed into hydroperoxides by LOX, which is consistent with the corresponding aroma products of the metabolic flux [16], and this is why LOX had been traditionally used as a marker for flavor [96]. *PpaLOX2.2* was identified in peach and may be required while fruit ripening during storage [95]. The expression of *LOX2* is higher at the beginning of the ripening process, decreasing until complete fruit maturity in all the cultivars analyzed. *LOX2* is a candidate gene for transforming aldehyde from fatty acids by *β*-oxidation, yielding the precursors for the terpene pathway [97]. Furthermore, due to its implication in the fruit ripening process and senescence, *LOX2* is a good candidate for monitoring the ripening process in apricot fruits, highly expressed at the beginning of the ripening process before the change of color.

## 4. Conclusions

Results showed that the construction of multiple linear regression (MLR) models for anthocyanins and carotenoids content allowed the identification of the genes having the greatest influence on the response variables (i.e., anthocyanins and carotenoids content). These genes are key genes during metabolite biosynthesis or ripening process regulation, responsible for the final fruit phenotype. The MLR model analyzed the influence of each gene in an integrated manner, which is a proper method for the prospection of phenotypic characters controlled by multiple genes. As a result, two MLR models with satisfactory explanation percentages were built, although the model for anthocyanins content did not comply with all the statistical hypotheses for inference purposes. The model for anthocyanins content included the gene *CHALCONE SYNTHASE*, involved in anthocyanin biosynthesis, described as key genes for anthocyanins content; while *CAROTENOID CLEAVAGE DIOXYGENSASE 4*, responsible for carotenoids degradation and which expression is linked to light yellow flesh fruit color, together with *ZETA-CAROTENE DESATURASE*, are validated as key genes for carotenoids content and fruit color. In relation to the ripening process, *LIPOXYGENASE 2* and *MADS-BOX TF* are proposed as candidate genes for monitoring the progression of the ripening process in apricot fruit. All these genes could be applied as RNA markers to monitor ripening stage and estimate the anthocyanins and carotenoids content in apricot fruit in the ripening process.

## 5. Materials and Methods

### 5.1. Plant Material

The plant material consisted of nine apricot (*Prunus armeniaca* L.) genotypes including ‘Bergeron’, ‘Cebasred’, ‘Currot’, ‘Dorada’, ‘Estrella’, ‘Goldrich’, ‘Moniquí’, ‘Valorange’ and ‘Z 108-38′. This set of apricot cultivars has wide genetic as well as phenotypic diversity, including different fruit quality traits in terms of ripening date, fruit color or ethylene production (Table 2) (personal communication from Dr David Ruiz, leader at CEBAS-CSIC apricot breeding program).

### 5.2. Evaluation of Fruit Quality Traits

Ten fruits from each cultivar were collected at three different ripening stages before stone hardening based on their skin ground color and firmness: green fruit (Stage A), during color change (Stage B) and at physiological ripening (Stage C) (Figure 13). Ripening date and fruit quality traits at physiological ripening were analyzed including physical characterization (skin color, flesh color and firmness), biochemical compounds (soluble solids content and titratable acidity). Total chlorophylls, carotenoids and anthocyanins content was evaluated at three different ripening stages based on their skin ground color and firmness: green fruit (Stage A), during color change (Stage B) and at physiological ripening (Stage C) by spectrophotometric techniques from edible fruit portions (skin and flesh). Fruit quality evaluation protocols were previously described by García-Gómez et al. [22]. Anthocyanins content was determined by a pH differential method from a pool of ten fruit (edible portion) at three ripening stages (green fruit, during color change and at physiological ripening) of apricot genotypes. Three biological replicates for each ripening stage and genotype were analyzed. Total anthocyanins were measured according to the modified method described as follows [98]. Two sample dilutions were prepared, one for pH 1.0 using potassium chloride buffer 0.03 M and the other for pH 4.5 using sodium acetate buffer 0.4 M. Samples were diluted 10 times to a final volume of 2 mL. The absorbance of each sample was measured at 520 nm. The samples had no haze or sediment and correction at 700 nm was omitted. Total anthocyanins content was referred to cyanidin-3-O-rutinoside equivalents [20].

### 5.3. Gene Expression Analysis through RT-qPCR

Samples of ten fruits (edible portions) from nine assayed cultivars were collected for RNA extraction at three different ripening stages based on their skin ground color and firmness: green fruit (Stage A), during color change (Stage B) and at physiological ripening (Stage C). Total RNA extraction and qRT-PCR was performed as by Garcia Gómez et al. [22,99]. To evaluate the expression pattern, 25 genes located on QTLs, DEG in RNA-Seq linked or described in the bibliography related to fruit color and ripening progression were evaluated [1,22,99,100]. The monitored genes included genes involved in the regulation of ripening progress including *COMMON PLANT REGULATORY FACTOR 1-LIKE* (*bZIP*), *LIPOXYGENASE 2* (*LOX2*), *MADS BOX TRANSCRIPTION FACTOR* (*MADS-box*), *NAC DOMAIN-CONTAINING PROTEIN 72-LIKE* (*NAC*), *PSBP DOMAIN CONTAINING PROTEIN 6, CHLOROPLASTIC* (*psbP1*), *PROBABLE SERINE/THREONINE-PROTEIN KINASE PBL16* (*WRKY*) and *SERINE/THREONINE–PROTEIN KINASE EDR1-LIKE* (*EDR1*); anthocyanins biosynthesis including *ANTHOCYANIDIN SYNTHASE* (*ANS*), *BASIC HELIX-LOOP-HELIX DNA-BINDING DOMAIN* (*bHLH*), *TRANS-CINNAMATE 4-MONOOXYGENASE* (*C4H*), *CINNAMYL-ALCOHOL DEHYDROGENASE* (*CAD1*), *CHALCONE/STILBENE SYNTHASE* (*CHS*), *4-COUMARATE COA LIGASE* (*4CL*), *DIHYDROFLAVONOL 4-REDUCTASE* (*DFR*), *FLAVONOID 3′5′-HYDROXYLASE 1-LIKE* (*F3′5′H*), *MYB TRANSCRIPTION FACTOR* (*MYB10*), *UDP FLAVONOID 3-O-GLUCOSYLTRANSFERASE* (*UFGT*) and *CONSERVED WD40 REPEAT-CONTAINING PROTEIN* (*WD40*); carotenoids biosynthesis including *CAROTENOID CLEAVAGE DIOXYGENASE 1* (*CCD1*), *CAROTENOID CLEAVAGE DIOXYGENASE 4* (*CCD4*), *CAPSANTHIN/CAPSORUBIN SYNTHASE, CHROMOPLASTIC-LIKE* (*CrtL*), *1-DEOXY-D-XYLULOSE-5-PHOSPHATE SYNTHASE* (*DXS*) and *ZETA CAROTENE DESATURASE* (*ZDS*); and metabolite transporters including *GLUTATHIONE TRANSFERASE GST 23-LIKE* (*GST*) and *MULTIDRUG*
*AND TOXIN EFFLUX TRANSPORTERS (MATE)* (Appendix A).

### 5.4. Statistical Analysis and Visualization

Statistical analysis and multiple linear regression (MLR) models were performed with R version 3.5.1. The variable whose value is to be predicted is known as the dependent, and the ones whose known values are used for prediction are known as the independent (exploratory) variables. After a stepwise method based on selecting the best predictors with the Akaike Information Criterion (AIC), the best MLR model was obtained. Package FactoMineR version 1.42 [101] was used for descriptive statistics and Principal Components Analysis (PCA). Visualization of correlations was performed with the Package Corrplot version 0.84 [102]. All the plots were executed with the package ggplot2 version 3.3.5 [103].

## Figures and Tables

**Figure 1 ijms-23-04575-f001:**
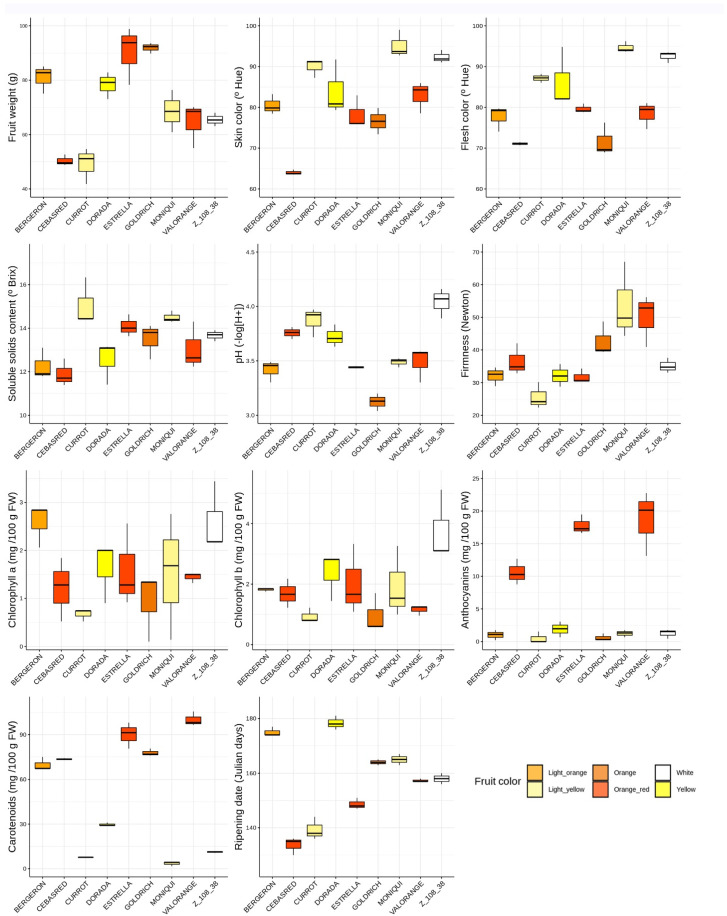
Phenological and fruit quality traits boxplots (fruit weight, skin color, flesh color, soluble solids content, acidity (pH), firmness, chlorophyll *a*, chlorophyll *b*, anthocyanins, carotenoids and ripening date) evaluation of the nine cultivars assayed, ‘Bergeron’, ‘Cebasred’, ‘Currot’, ‘Dorada’, ‘Estrella’, ‘Goldrich’, ‘Moniquí’, ‘Valorange’ and ‘Z 108-38′ at physiological ripening. Boxplots are filled with the fruit color of each cultivar.

**Figure 2 ijms-23-04575-f002:**
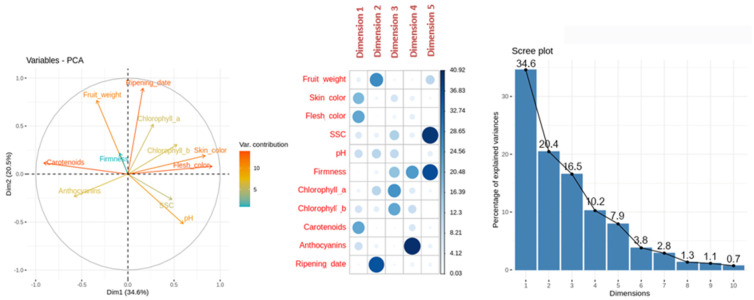
Principal Component Analysis (PCA). From left to right: representation of the variable contribution to dimension 1 and 2, variable contribution and dimension correlation with the first five dimensions and percentage of explained variance for the first ten dimensions performed over phenological and quality traits (fruit weight, skin color, flesh color, soluble solids content (SSC), acidity (pH), firmness, chlorophyll *a*, chlorophyll *b*, anthocyanins, carotenoids and ripening date) evaluated in apricot fruit at physiological ripening. The cultivars evaluated were ‘Bergeron’, ‘Cebasred’, ‘Currot’, ‘Dorada’, ‘Estrella’, ‘Goldrich’, ‘Moniquí’, ‘Valorange’ and ‘Z108-38′.

**Figure 3 ijms-23-04575-f003:**
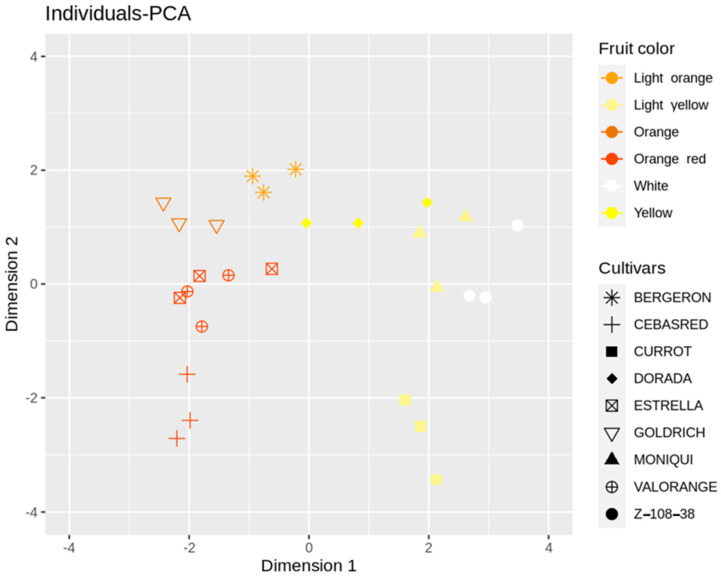
Principal Component Analysis (PCA) performed over the phenological and fruit quality traits (fruit weight, skin color, flesh color, soluble solids content (SSC), acidity (pH), anthocyanins and carotenoids content and ripening date (RD)) of nine cultivars ‘Bergeron’, ‘Cebasred’, ‘Currot’, ‘Dorada’, ‘Estrella’, ‘Goldrich’, ‘Moniquí’, ‘Valorange’ and ‘Z 108-38′ at physiological ripening. Symbols represent each cultivar and are colored as the real fruit color previously described. This color legend will be used for all bar plots in this study.

**Figure 4 ijms-23-04575-f004:**
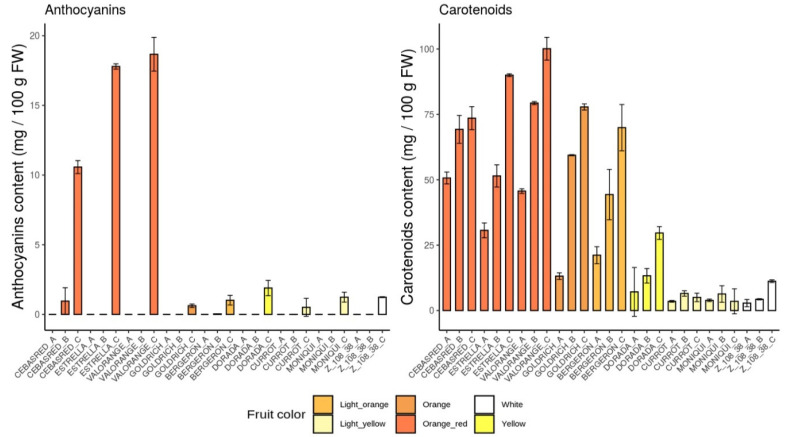
Anthocyanins and carotenoids content bar plots of the nine cultivars ‘Bergeron’, ‘Cebasred’, ‘Currot’, ‘Dorada’, ‘Estrella’, ‘Goldrich’, ‘Moniquí’, ‘Valorange’ and ‘Z 108-38′ at three different ripening stages based on their skin ground color and firmness: green fruit (Sample A), during color change (Sample B) and at physiological ripening (Sample C).

**Figure 5 ijms-23-04575-f005:**
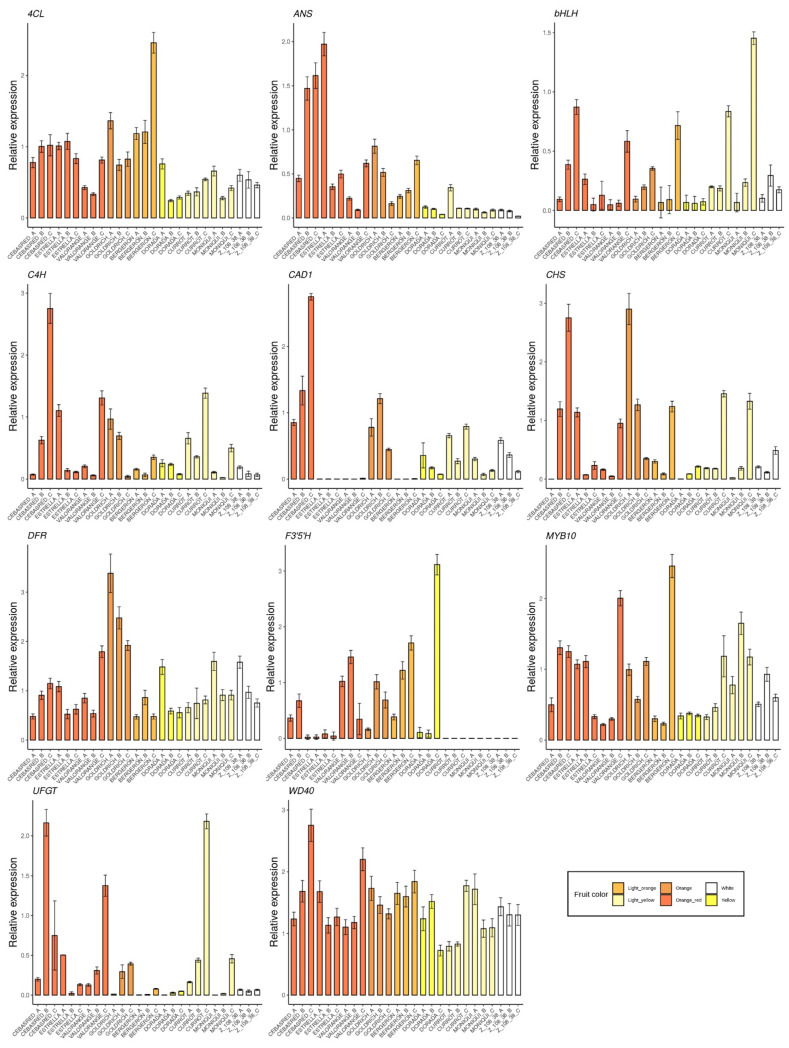
Expression dynamics barplots of the genes involved in anthocyanin biosynthesis *(4CL, ANS, bHLH, C4H, CAD1, CHS, DFR, F3′5′H, MYB10, UFGT* and *WD40*) in the nine cultivars ‘Bergeron’, ‘Cebasred’, ‘Currot’, ‘Dorada’, ‘Estrella’, ‘Goldrich’, ‘Moniquí’, ‘Valorange’ and ‘Z 108-38′ at three different ripening stages based on their skin ground color and firmness: green fruit (_A), during color change (_B) and at physiological ripening (_C).

**Figure 6 ijms-23-04575-f006:**
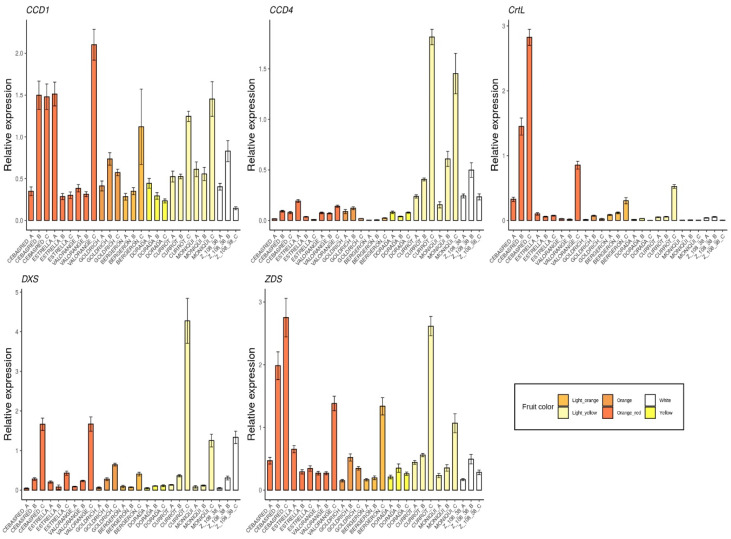
Expression dynamics barplots of the genes involved in carotenoids biosynthesis and degradation (*CCD1, CCD4, CrtL, DXS* and *ZDS*) in the nine cultivars ‘Bergeron’, ‘Cebasred’, ‘Currot’, ‘Dorada’, ‘Estrella’, ‘Goldrich’, ‘Moniquí’, ‘Valorange’ and ‘Z 108-38′ at three different ripening stages based on their skin ground color and firmness: green fruit (_A), during color change (_B) and at physiological ripening (_C).

**Figure 7 ijms-23-04575-f007:**
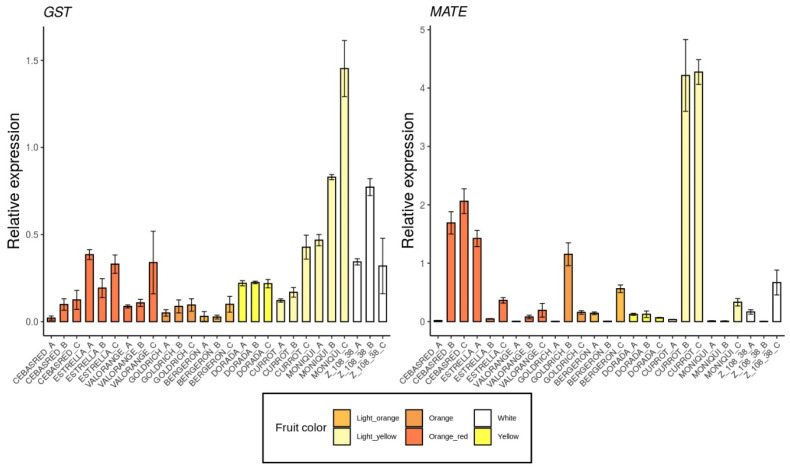
Expression dynamics barplots of the genes described as metabolites transporters *GST* and *MATE*, in the nine cultivars ‘Bergeron’, ‘Cebasred’, ‘Currot’, ‘Dorada’, ‘Estrella’, ‘Goldrich’, ‘Moniquí’, ‘Valorange’ and ‘Z 108-38′ at three different ripening stages based on their skin ground color and firmness: green fruit (_A), during color change (_B) and at physiological ripening (_C).

**Figure 8 ijms-23-04575-f008:**
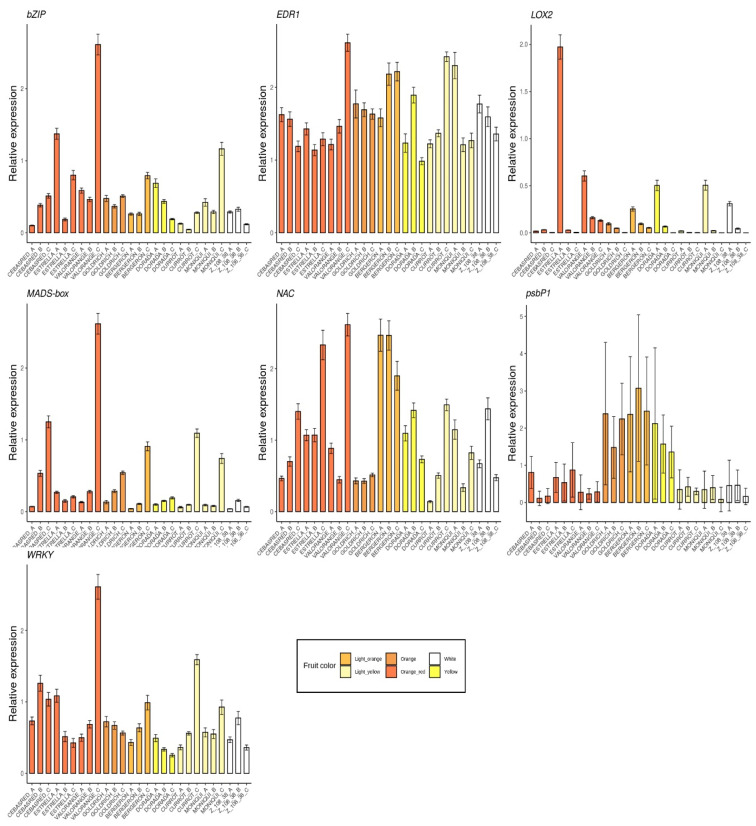
Barplots of the expression dynamics of others genes involved in the regulation of ripening progress, as *bZIP, EDR1, LOX2, MADS-box, NAC, psbP1* and *WRKY*, in the nine cultivars ‘Bergeron’, ‘Cebasred’, ‘Currot’, ‘Dorada’, ‘Estrella’, ‘Goldrich’, ‘Moniquí’, ‘Valorange’ and ‘Z 108-38′ at three different ripening stages based on their skin ground color and firmness: green fruit (_A), during color change (_B) and at physiological ripening (_C).

**Figure 9 ijms-23-04575-f009:**
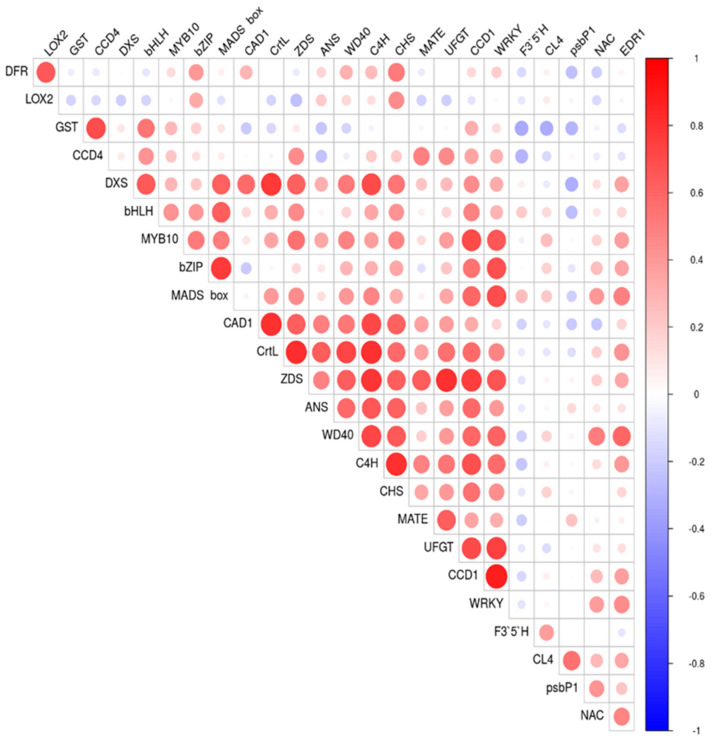
Pearson correlation plot for the relative gene expression evaluated by using RT-qPCR of genes involved in anthocyanins and carotenoids biosynthesis and ripening process progression in the nine cultivars ‘Bergeron’, ‘Cebasred’, ‘Currot’, ‘Dorada’, ‘Estrella’, ‘Goldrich’, ‘Moniquí’, ‘Valorange’ and ‘Z108-38′ at three ripening stages green fruit, during color change and at physiological ripening. The assayed genes included *DFR, LOX2, GST, CCD4, DXS, bHLH, MYB10, bZIP, MADS-box, CAD1, CrtL, ZDS, ANS, WD40, C4H, CHS, MATE, UFGT, CCD1, WRKY, F3′5′H, CL4, psbP1, NAC* and *EDR1*.

**Figure 10 ijms-23-04575-f010:**
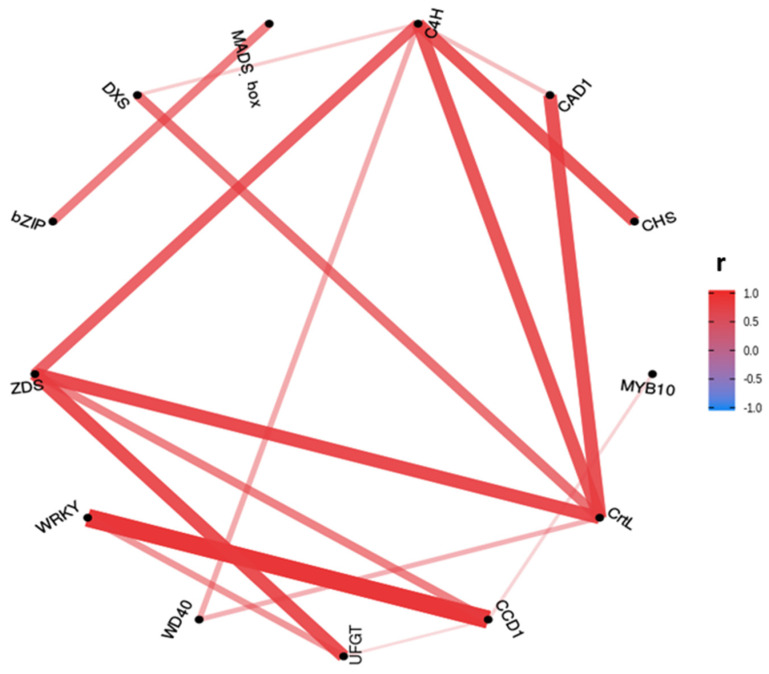
Pearson correlation (r coefficient) network for relative expression by RT-qPCR of genes involved in anthocyanins and carotenoids biosynthesis with correlation values over 0.7. The plotted genes were *C4H, CAD1, CHS, MYB10, CrtL, CCD1, UFGT, WD40, WRKY, ZDS, bZIP, DXS* and *MADS-box*.

**Figure 11 ijms-23-04575-f011:**
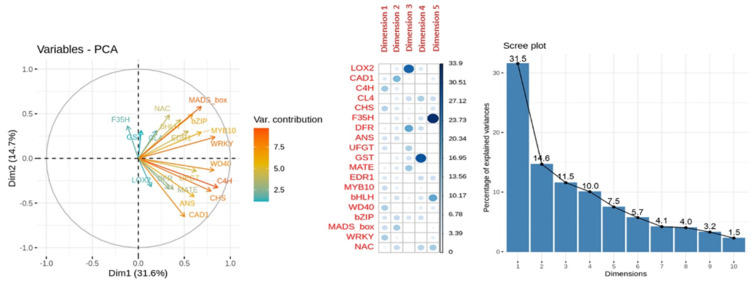
Principal Component Analysis (PCA) of relative expression from anthocyanins content MLR model. From left to right: representation of the variable contribution to dimensions 1 (Dim1) and 2 (Dim 2), variable contribution and dimension correlation with the first five dimensions and percentage of explained variance of the first ten dimensions performed over the relative expression of the genes involved in anthocyanins biosynthesis (*ANS, bHLH, bZIP, C4H, CAD1, CHS, CL4, DFR, EDR1, F3′5′H, GST, MATE, MADS-box, MYB10, NAC, LOX2, UFGT, WD40* and *WRKY*) evaluated in apricot fruit during the ripening progression in the nine cultivars ‘Bergeron’, ‘Cebasred’, ‘Currot’, ‘Dorada’, ‘Estrella’, ‘Goldrich’, ‘Moniquí’, ‘Valorange’ and ‘Z108-38′ at three ripening stages green fruit, during color change and at physiological ripening.

**Figure 12 ijms-23-04575-f012:**
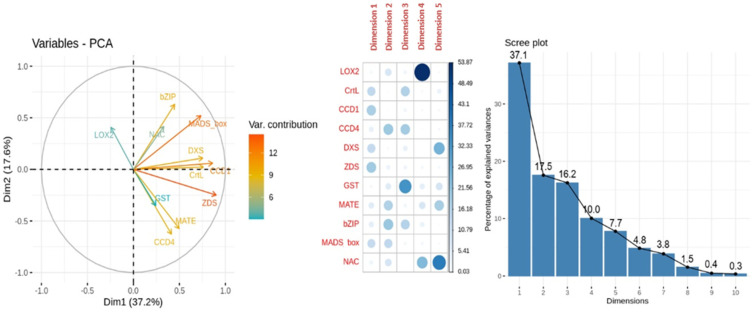
Principal Component Analysis (PCA) of relative expression from carotenoids content MLR model. From left to right: representation of the variable contribution to dimensions 1 (Dim1) and 2 (Dim2), variable contribution and dimension correlation with the first five dimensions and percentage of explained variance of the first ten dimensions performed over the relative expression of the genes involved in carotenoid biosynthesis (*bZIP, CCD1, CCD4, CrtL, DXS, GST, MATE, MADS-box, NAC, LOX2* and *ZDS*) evaluated in apricot fruit during the ripening progression in the nine cultivars ‘Bergeron’, ‘Cebasred’, ‘Currot’, ‘Dorada’, ‘Estrella’, ‘Goldrich’, ‘Moniquí’, ‘Valorange’ and ‘Z108-38′ at three ripening stages green fruit, during color change and at physiological ripening.

**Figure 13 ijms-23-04575-f013:**
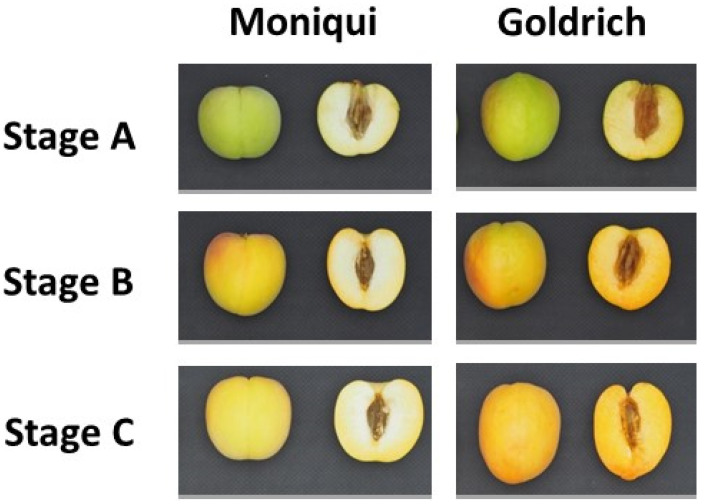
Ripening stages before stone hardening based on their skin ground color and firmness assayed in this study in the light yellow cultivar ’Moniquí’ and the orange cultivar ‘Goldrich’: green fruit (**Stage A**), during color change (**Stage B**) and at physiological ripening (**Stage C**).

**Table 1 ijms-23-04575-t001:** Descriptive statistics (mean and standard deviation (SD)) of phenological ripening date and quality traits (fruit weight, skin color, flesh color, soluble solids content (SSC), acidity (pH), firmness, chlorophyll *a*, chlorophyll *b,* anthocyanins and carotenoids content) analyzed in nine apricot cultivar cultivars ‘Bergeron’, ‘Cebasred’, ‘Currot’, ‘Dorada’, ‘Estrella’, ‘Goldrich’, ‘Moniquí’, ‘Valorange’ and ‘Z 108-38′ at physiological ripening.

Cultivar	Trait	Mean	SD	Cultivar	Trait	Mean	SD	Cultivar	Quality Trait	Mean	SD
‘Bergeron’	Fruit weight	81.0	5.23	‘Dorada’	Fruit weight	78.4	4.95	‘Moniquí’	Fruit weight	68.6	7.75
	Skin color	80.5	2.45		Skin color	84.0	6.75		Skin color	95.2	3.38
	Flesh color	77.7	3.13		Flesh color	86.3	7.36		Flesh color	94.7	1.37
	SSC	12.3	0.72		SSC	12.5	0.97		SSC	14.5	0.265
	pH	3.42	0.09		pH	3.72	0.10		pH	3.49	0.041
	Firmness	32.0	2.86		Firmness	32.1	3.44		Firmness	53.7	11.8
	Chlorophylls *a*	2.58	0.45		Chlorophylls *a*	1.63	0.63		Chlorophylls *a*	1.5	1.32
	Chlorophylls *b*	1.82	0.05		Chlorophylls *b*	2.36	0.79		Chlorophylls *b*	1.93	1.18
	Anthocyanins	1.02	0.34		Anthocyanins	1.90	0.54		Anthocyanins	1.24	0.35
	Carotenoids	69.9	4.40		Carotenoids	29.6	1.18		Carotenoids	3.53	1.62
	Ripening date	175	1.73		Ripening date	178	2.52		Ripening date	165	2
‘Cebasred’	Fruit weight	50.4	2.03	‘Estrella’	Fruit weight	90.3	10.7	‘Valorange’	Fruit weight	64.6	8.28
	Skin color	64.0	0.62		Skin color	78.3	4.02		Skin color	82.9	3.88
	Flesh color	71.1	0.35		Flesh color	79.6	1.13		Flesh color	78.4	3.32
	SSC	11.9	0.62		SSC	14.1	0.50		SSC	13.1	1.10
	pH	3.76	0.05		pH	3.44	0.01		pH	3.49	0.15
	Firmness	36.5	4.83		Firmness	31.8	2.16		Firmness	50.0	8.03
	Chlorophylls *a*	1.21	0.66		Chlorophylls *a*	1.59	0.86		Chlorophylls *a*	1.44	0.10
	Chlorophylls *b*	1.69	0.47		Chlorophylls *b*	2.03	1.16		Chlorophylls *b*	1.15	0.16
	Anthocyanins	10.6	0.46		Anthocyanins	17.8	0.18		Anthocyanins	18.7	1.21
	Carotenoids	73.6	0.51		Carotenoids	90.0	8.85		Carotenoids	100	4.80
	Ripening date	134	3.21		Ripening date	149	2.08		Ripening date	157	0.57
‘Currot’	Fruit weight	49.2	6.62	‘Goldrich’	Fruit weight	91.9	1.94	‘Z108-38′	Fruit weight	65.5	2.49
	Skin color	89.9	2.33		Skin color	76.6	3.22		Skin color	92.3	1.57
	Flesh color	87.2	1.07		Flesh color	71.6	4.03		Flesh color	92.5	1.39
	SSC	15.1	1.10		SSC	13.5	0.81		SSC	13.7	0.25
	pH	3.87	0.13		pH	3.12	0.08		pH	4.04	0.13
	Firmness	25.6	4.09		Firmness	42.7	5.21		Firmness	35.1	2.30
	Chlorophylls *a*	0.66	0.12		Chlorophylls *a*	0.92	0.71		Chlorophylls *a*	2.60	0.03
	Chlorophylls *b*	0.93	0.24		Chlorophylls *b*	0.97	0.63		Chlorophylls *b*	3.78	1.17
	Anthocyanins	0.51	0.64		Anthocyanins	0.62	0.12		Anthocyanins	1.24	0.70
	Carotenoids	7.58	0.14		Carotenoids	77.8	2.46		Carotenoids	11.2	0.50
	Ripening date	139	4.16		Ripening date	164	1		Ripening date	15	2

**Table 2 ijms-23-04575-t002:** Apricot cultivars assayed.

Cultivar	Pedigree	Origin	Ripening	Fruit Color	Ethylene Production
Bergeron‘	Unknown	France	Very late	Light orange	Very low
‘Cebasred‘	‘5–26‘ × ‘Colorado‘	Spain	Very early	Orange–Red	Low
‘Currot‘	Unknown	Spain	Very early	Light yellow	High
‘Dorada‘	‘Bergeron‘ × ‘Moniquí‘	Spain	Late	Yellow	Low
‘Estrella‘	‘Orange–Red‘ × ‘Z211-18‘	Spain	Medium	Orange–Red	Medium
‘Goldrich‘	‘Sunglo‘ × ‘Perfection‘	USA	Late	Orange	Very low
‘Moniquí‘	Unknown	Spain	Medium	Light yellow	High
‘Valorange‘	‘Orange–Red‘ × ‘Currot‘	Spain	Medium	Orange–Red	Low
‘Z108-38‘	‘Gitano’ × ‘Pepito del Rubio’	Spain	Late	White	High

## Data Availability

Not applicable.

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
