# Peer review of "Monitoring Apricot (Prunus armeniaca L.) Ripening Progression through Candidate Gene Expression Analysis"

_ijms, 2022, doi:10.3390/ijms23094575_

Round 1

Reviewer 1 Report

Reviewer Response is attached as PDF

Author Response

April 18, 2022

Dear Ms. Manuela Marcus, Assistant Editor of International Journal of Molecular Sciences,

please find enclosed the manuscript ijms-ijms-1678625-R1 entitled “Monitoring apricot (Prunus armeniaca L.) ripening progression through candidate gene expression analysis” which is the revised version of the manuscript which we would like to publish in your journal.

According with the suggestions of the Reviewer 1 we have revised the manuscript incorporating the proposed revisions indicating these revisions with the control of changes of the WORD document.

We deeply appreciate the efforts of the reviewer in the improvement of the manuscript for a future publication.

Regarding reviewer's 1 comments (R1):

 R1: García-Gomez et al. (2022) describe an interesting study investigating the biomolecular background of ripening processes in Apricot cultivars. The horticultural production of fruits, among them Apricots, is an important industrial factor for many countries in particular around the Mediterranean basin. Besides alternative pest management and sustainable production approaches, measures to achieve homogeneous fruit ripening are of primary interest for fruit growers. Although the manuscript is well written and its strong molecular background, I am not sure if the International Journal of Molecular Sciences is the correct journal`s choice, because most of its manuscript deal with medicine and fundamental biological issues, rather than agri-/horticultural topics. The chosen experimental design is sounding to me. Due to the good quality of the submitted manuscript, I have few things to add in the

following.

Authors: We agree and thank the Reviewer 1 for their valuable comments about the revision of this work. In addition, all the suggestions and revisions of the reviewer have been incorporated indicating these revisions with the “Track Changes” of the WORD document.

 R1: Abstract

- Shorten the introduction section

- Include concrete key results

- Change “species” to “cultivars” line 1

 Authors: The introduction section of the abstract was shorten and the concrete results about which genes were key into pigments content were already included.

R1: Results

- If I counted correctly, there are 11 main figures with a total of 49 subfigures in the manuscript, whereas in the supplementary material (only) four subfigures are presented. All of them are of good quality, but in my opinion a significant reduction of figures would be a benefit for the manuscript; thereby focusing the research. As an idea the authors could use heat maps for illustrate overall gene expression lelves, take insignificant genes out or move them to the supplements.

- Introduce line spacing after Table 1

Authors: Although the proposition of plot the main results obtained from gene expression analysis as an heat map is very attractive, was decided to plot as histograms for all the genes initially included in the model because of the possibility to include the standard deviation bars in the plot, and the easily comparison between bars height in the different varieties and stages, which is harder when represented as colors scales. Besides, each variety bars were filled with their fruit color, giving us this important information to interpret the plot in one view.

R1: Material and Methods

- Include the taxonomical name of apricots in the beginning of the M&M section e.g. Line 781

- Skip “very” line 784

- Is there a reference for the ethylene production (and sensitivity) for Table 2?

- Add references for the anthocyanin, chlorophyll, color and firmness methodologies (lines 793 and following).

- The gene names in lines 819 and following should not be included in the flow text. A supplementary table would increase the readability and authors could add more methodological information e.g. about primer sequenes, efficiencies, R2

Authors: The apricot taxonomical name was included in the beginning of the M&M section.

The description of main characteristics of apricot varieties selected for the present study was facilitated by Dr David Ruiz, the leader of CEBAS-CSIC apricot breeding program.

The protocols descriptions for all the quality traits evaluated in the present study were previously described in García-Gómez et al. [22] (reference included in text and bibliography).

Following the recommendation of the reviewer, a supplementary table was included with the description of gene analyzed and its primer sequences.

R1: Discussion and conclusion

A main focus of the manuscript is to connect alterations of gene expression levels to ripening processes. Transcriptomics are undoubtedly one of the major factors as seen in other fruit ripening processes (e.g. for grapevine). However other factors that might interfere the hypotheses should be critically discussed. This would involve tissue specifities, enzyme activity, availability of enzymatic substrates, etc.. Likewise the conclusion should be relativize the key results and have a clear focus on Apricots

Authors: As the reviewer indicates, the results obtained were relativize due to the experimental design and other condition which could affect the biosynthesis and accumulation of pigment compounds in apricot fruit during ripening process, as was exposed the importance of regulatory mechanism and environmental influence of incidence light and temperature and dilution effect due sampling the whole edible fruit for anthocyanin content estimation, the low sensibility of the method used for quantifying anthocyanins content, the presence of interactions between the predictors or the not inclusion of important predictors during the model selection step in carotenoids content.

We deeply appreciate the efforts of the reviewer in the improvement of the manuscript for a future publication.

Yours faithfully,

Dr. Pedro Martínez-Gómez

CEBAS-CSIC, Murcia (Spain)

Reviewer 2 Report

The paper by Garcia-Gomez and colleagues represents a sound example of using expression data to monitor ripening in fruits, with promising predictive features of the MLR models set up in the analysis. The scientific soundness is high and results are clearly presented. The discussion also allows for interesting hints. The paper should be accepted after very minor improvements.

Given the importance of MADS-domain transcription factors in fruit development and ripening, it would be beneficial to the paper if the authors added a sequence comparison to identify the closest-known MADS protein. Is it a SEPALLATA protein, or am i misunderstanding the discussion?

Finally, I would only change the design of table 1, in order to reflect the structure of the actual dataset used by the R packages used and, with that, improve its readability also for the audience. My suggestion would be using rows for the apricot varieties and columns for the measured traits, something like this: 

  Fruit weight Skin colour ...
Bergeron 81 +/- 5.23 ... ...
Morada ... ... ...
... ... ... ...

Author Response

April 18, 2022

Dear Ms. Manuela Marcus, Assistant Editor of International Journal of Molecular Sciences,

please find enclosed the manuscript ijms-ijms-1678625-R1 entitled “Monitoring apricot (Prunus armeniaca L.) ripening progression through candidate gene expression analysis” which is the revised version of the manuscript which we would like to publish in your journal.

According with the suggestions of the Reviewer 2 we have revised the manuscript incorporating the proposed revisions indicating these revisions with the control of changes of the WORD document.

We deeply appreciate the efforts of the reviewer in the improvement of the manuscript for a future publication.

Regarding reviewer's 2 comments (R2):

 R2: The paper by Garcia-Gomez and colleagues represents a sound example of using expression data to monitor ripening in fruits, with promising predictive features of the MLR models set up in the analysis. The scientific soundness is high and results are clearly presented. The discussion also allows for interesting hints. The paper should be accepted after very minor improvements.

Given the importance of MADS-domain transcription factors in fruit development and ripening, it would be beneficial to the paper if the authors added a sequence comparison to identify the closest-known MADS protein. Is it a SEPALLATA protein, or am i misunderstanding the discussion?

Finally, I would only change the design of table 1, in order to reflect the structure of the actual dataset used by the R packages used and, with that, improve its readability also for the audience. My suggestion would be using rows for the apricot varieties and columns for the measured traits, something like this:

                                                     Fruit weight          Skin colour      ...

Bergeron                                      81 +/- 5.23 ...         ...

Morada                                        ...   ...         ...

 Authors: We agree and thank the Reviewer 2 for their valuable comments about the revision of this work. In addition, all the suggestions and revisions of the reviewer have been incorporated indicating these revisions with the “Track Changes” of the WORD document.

As the reviewer suggest, will be of great interest to analyse the sequence of MADS-box gene found up-regulated at the end of ripening process to identify the closes-known MADS protein. But the result of this analys was mainly focused on the differential expression of genes involved in pigment content during the ripening process in apricot to identify the key genes responsible for fruit color.

As those data were analysed in R, the suggestion of format this table as R table with the samples in the rows and the variables in columns is very attractive. Finally was decided to design in this way to highlighted the difference between apricot varieties.

We deeply appreciate the efforts of the reviewer in the improvement of the manuscript for a future publication.

Yours faithfully,

Dr. Pedro Martínez-Gómez

CEBAS-CSIC, Murcia (Spain)
